# Access to radiotherapy in improving gastric cancer care quality and equality
Minmin Wang [1,2], Kepei Huang[1,2], Xiaohan Fan[3], Jia Wang[4], Yinzi Jin [1,2] ✉ & Zhi-Jie Zheng[1,2]

## Abstract

**Background** Quality health services could improve patient outcomes and prognosis. Gastric cancer care was of great disparity across genders. Disparities within radiotherapy units could impact gastric cancer care, potentially exacerbating gender-based inequalities. **Methods** We retrieved the disease burden data from Global Burden of Disease 2019. A quality of care index was constructed by applying principal component analysis techniques. The disparity of gastric cancer care across genders was described, and the association of access to radiotherapy with gastric cancer care as well as gender disparity was explored. **Results** Males receive better quality of gastric cancer care than females, and this gender disparity is widening in middle-low socio-development regions. A positive correlation emerges between the density of radiotherapy facilities and an elevated quality of care, and reduced gender-based disparities. **Conclusions** The association between robust radiotherapy access, improved gastric cancer QCI, and reduced gender-based disparities spotlights the imperative of fortifying radiotherapy infrastructure within areas and populations in greatest need.

## Plain Language Summary

Gastric cancer is a prevalent disease which ranked fifth for incidence and fourth for mortality among all cancer types globally. Great differences in the quality of gastric cancer care have been reported across regions, countries and genders. Accessibility to essential treatment, especially radiotherapy units used in treatment, could impact the quality of gastric cancer care. We studied the role of health technologies in promoting gastric cancer care quality and equality. Our results identified an association between radiotherapy access and improved quality of gastric cancer care and reduced gender-based disparities. Results of this study highlighted the importance of increasing access to radiotherapy treatment in improving global health equality.

Gastric cancer is a type of malignant tumor originating from the gastric mucosal epithelium. In 2020, gastric cancer ranked fifth for incidence and fourth for mortality among all cancer types globally, with more than 1 million new cases and about 769,000 new deaths, accounting for 1 in 13 global deaths[1,2]. There are regional differences in the incidence and mortality of gastric cancer, with more than 60% of gastric cancer cases occurring in East and Southeast Asia. East Asia has the highest regional gastric cancer mortality rate (15.9 deaths per 100,000 population), followed by Central and Eastern Europe and South America[3].

Providing quality health services is key to achieving universal health coverage[4], and is an essential measure to improve patients' quality of life and reduce the cancer burden[5,6]. Previous studies have estimated the global quality of gastric cancer care, and reported great disparity across regions, countries and genders[7], which limited the progress to improve the quality of life in gastric cancer patients as well as to improve the global health equity. However, neither the mechanism behind the disparity of quality of gastric

cancer care across regions, countries and genders was fully understood, nor the improving strategies were informed.

Access to essential treatment infrastructure, especially the radiotherapy units, could impact the quality of gastric cancer care and reduce gender-based disparity. Arranging access to and coordinating appropriate treatment was among the core basic of delivering cancer care. In 2022, the "Setting Up a Cancer Centre: A WHO-IAEA Framework" proposed a comprehensive blueprint for establishing and enhancing of cancer centers. This framework emphasized crucial components such as infrastructure, human resources, and essential equipment as fundamental prerequisites for rendering indispensable services[8]. Most patients with gastric cancer are locally advanced and require multimodal treatment[9], in which radiotherapy is emphasized. It is generally accepted that adjuvant and neoadjuvant therapies, such as adjuvant chemoradiotherapy and preoperative chemoradiotherapy, can improve the disease-free and overall survival of patients with locally advanced gastric cancer who have undergone adequate and

[1]Department of Global Health, School of Public Health, Peking University, Beijing, China. [2]Institute for Global Health and Development, Peking University, Beijing, China. [3]Key Laboratory of Carcinogenesis and Translational Research (Ministry of Education/Beijing), Department of Cancer Epidemiology, Peking University Cancer Hospital & Institute, Beijing, China. [4]Key Laboratory of Carcinogenesis and Translational Research (Ministry of Education/Beijing), Peking University Cancer Hospital & Institute, Beijing, China. ✉e-mail: yzjin@bjmu.edu.cn

complete surgical resection[10]. With these therapies, the 5-year overall survival rate can be increased by 10–15%[11].

Thus, we postulate that disparities within radiotherapy units could impact gastric cancer care, potentially exacerbating gender-based inequalities. This study aims to explore the mechanism of the disparity of gastric cancer care across regions, countries, and genders, by conducting an association analysis to identify the role of health technologies and health workforce in promoting gastric cancer quality and equality.

## Methods

### Data sources
The Global Burden of Disease (GBD) study assessed the disease burden of 369 different diseases and 87 risk factors in 204 countries, 21 regions, and seven super-regions, aiming to provide a comprehensive and comparable global health assessment[12,13]. GBD methodology has been described previously. Briefly, the GBD estimation process is based on identifying multiple relevant data sources for each disease including censuses, household surveys, civil registration and vital statistics, disease registries, health service use, air pollution monitors, satellite imaging, disease notifications, and other sources. Cause-specific death rates and cause fractions were calculated using the Cause of Death Ensemble model and spatiotemporal Gaussian process regression. Cause-specific deaths were adjusted to match the total all-cause deaths calculated as part of the GBD population, fertility, and mortality estimates. Deaths were multiplied by standard life expectancy at each age to calculate years of life lost (YLLs). A Bayesian meta-regression modeling tool, DisMod-MR 2.1, was used to ensure consistency between incidence, prevalence, remission, excess mortality, and cause-specific mortality for most causes. Prevalence estimates were multiplied by disability weights for mutually exclusive sequelae of diseases and injuries to calculate years lived with disability (YLDs). In this study, we searched the GBD 2019 database and collected data on gastric cancer incidence, prevalence, mortality, YLLs, YLDs, and disability-adjusted life-years (DALYs) from 1990 to 2019. Gastric cancer was defined according to the International Classification of Diseases Tenth Revision (ICD-10); codes C16–C16.9, D00.2, D13.1, and D37.1 were recorded as new cases of gastric cancer in the GBD dataset[14].

Country-level health technologies and health workforce data was retrieved from WHO Global Health Observatory (GHO, https://www.who.int/data/gho). The GHO data repository provided WHO's statistics on priority health topics including mortality and burden of diseases, non-communicable diseases and risk factors, health systems, environmental health, violence and injuries and others in 194 Member States. The total density of radiotherapy units per million population and number of medical doctors per 10,000 population were used to represent the health technologies and health workforce, separately. In 2010, WHO launched a country survey on medical devices that allowed to identify the status of high cost medical devices in the Member States, including radiotherapy equipment, both linear accelerators and Cobalt-60. Similar survey was conducted in 2020–2021 update by collecting information directly from country focal points from ministries of health.

### Quality of Care Index
First, four secondary indicators were constructed based on the age-standardized rate retrieved from the GBD 2019 dataset, namely[1] the ratio of YLLs to YLDs[2], the ratio of DALYs to prevalence[3], mortality-to-incidence ratio, and[4] prevalence-to-incidence ratio.

Ratio of YLLs and YLDs = YLLs/YLDs[1]
Ratio of DALYs to prevalence = DALYs/Prevalence[2]
Mortality-to-incidence ratio = Mortality/Incidence[3]
Prevalence-to-incidence ratio = Prevalence/Incidence[4]

Second, the quality of care index (QCI) was constructed using principal component analysis (PCA) techniques based on the above four secondary indicators. PCA involves mathematical multivariate analysis to extract linear combinations as orthogonal components of specific indicators. Then the component that best describes the variance and variability in the data is denoted the QCI and allocated a score of 0–100. Higher scores on the QCI indicate a high quality of gastric cancer care[15,16].

In this study, we calculated the QCI and analyzed the changing trends in the QCI from 1990 to 2019 for males and females. The estimated annual percentage change (EAPC) and 95% confidence intervals (CI) were calculated using a linear regression model[17]: $y = \alpha + \beta x + \varepsilon$, where $y = \ln(QCI)$, $x$ = calendar year, and $\varepsilon$ = error term. The value of EAPC equals $100 \times (\exp(\beta) - 1)$ and its 95% CI is attainable in the regression model.

A gender difference ratio (GDR) was calculated to explore the disparity between men and women[18]. The GDR was defined as the ratio of the QCI score in women divided by that in men, with a GDR > 1 indicating better gastric cancer care in women compared with men.

The socio-demographic index (SDI) is a comprehensive indicator based on the education level, per capita income, and total fertility rate of individuals under the age of 25 years, which measures the overall development scale of a country. The SDI indicator was also extracted from the GBD 2019 dataset (https://ghdx.healthdata.org/record/ihme-data/gbd-2019-socio-demographic-index-sdi-1950-2019). In 2019, different countries were classified into five development levels according to the SDI, namely: low (<0.46), low-middle (0.46–0.60), middle (0.61–0.69), high-middle (0.70–0.81), and high (>0.81)[19].

### Statistical analysis
Association analysis was conducted to identify the role of radiotherapy units in improving quality of gastric cancer care. Detailed information of the variable definition and coding forms were displayed in Supplementary Table 1. Univariable regression model was applied with the log-transformed gender-specific QCI as the outcome variable and the total density of radiotherapy units per million population was used as independent variable, separately. Multivariable models added the universal health coverage and social development levels defined using SDI as adjusting variables[19]. Model 3 further added health workforce (number of medical doctors per 100,000 population) and infrastructure (hospital beds per 10,000 population) as covariates. The effect size was estimated with 95% confidence interval (CI). Then parallel association analysis was conducted with the absolute value of log-transformed GDR was the outcome variable to identify the role of health technologies and health workforce in reducing gender inequality.

All statistical analyses for this study were created using R v4.1.3 software (http://www.r-project.org/). All tests were two-sided, and $P$ values < 0.05 were considered statistically significant.

### Ethics approval and consent to participate
Patient consent was not required as we only utilized de-identified, publicly available datasets in our analysis, Likewise, we did not seek ethics approval from any ethics committee or authoritative body for the use of this data in the study objectives as it was not required.

### Inclusion & ethics statement
All collaborators of this study have fulfilled the criteria for authorship required by Nature Portfolio journals have been included as authors, as their participation was essential for the design and implementation of the study. Roles and responsibilities were agreed among collaborators ahead of the research. This research was not severely restricted or prohibited in the setting of the researchers, and does not result in stigmatization, incrimination, discrimination or personal risk to participants. Local and regional research relevant to our study was taken into account in citations.

### Reporting summary
Further information on research design is available in the Nature Portfolio Reporting Summary linked to this article.

**Table 1 | The gastric cancer QCI in 1990 and 2019, by regions and genders**

| Region | Female | | | Male | | |
|---|---|---|---|---|---|---|
| | QCI in 2019 | QCI in 1990 | EAPC (95% CI) | QCI in 2019 | QCI in 1990 | EAPC (95% CI) |
| Global | 54.14 | 34.66 | 1.65 (1.57, 1.74) | 67.20 | 41.49 | 1.80 (1.70, 1.90) |
| High SDI | 86.00 | 70.12 | 0.72 (0.65, 0.80) | 90.50 | 76.07 | 0.62 (0.55, 0.68) |
| High-middle SDI | 58.90 | 26.60 | 3.17 (3.02, 3.33) | 68.93 | 30.82 | 3.21 (3.05, 3.37) |
| Middle SDI | 49.94 | 17.74 | 3.88 (3.75, 4.01) | 64.88 | 25.11 | 3.65 (3.49, 3.81) |
| Low-middle SDI | 21.36 | 12.31 | 1.92 (1.86, 1.97) | 33.35 | 17.96 | 2.26 (2.12, 2.39) |
| Low SDI | 13.31 | 10.29 | 0.98 (0.92, 1.05) | 17.90 | 14.97 | 0.67 (0.62, 0.71) |

*CI* confidential interval, *EAPC* estimated annual percentage change, *QCI* quality of cancer care, *SDI* sociodemographic index.

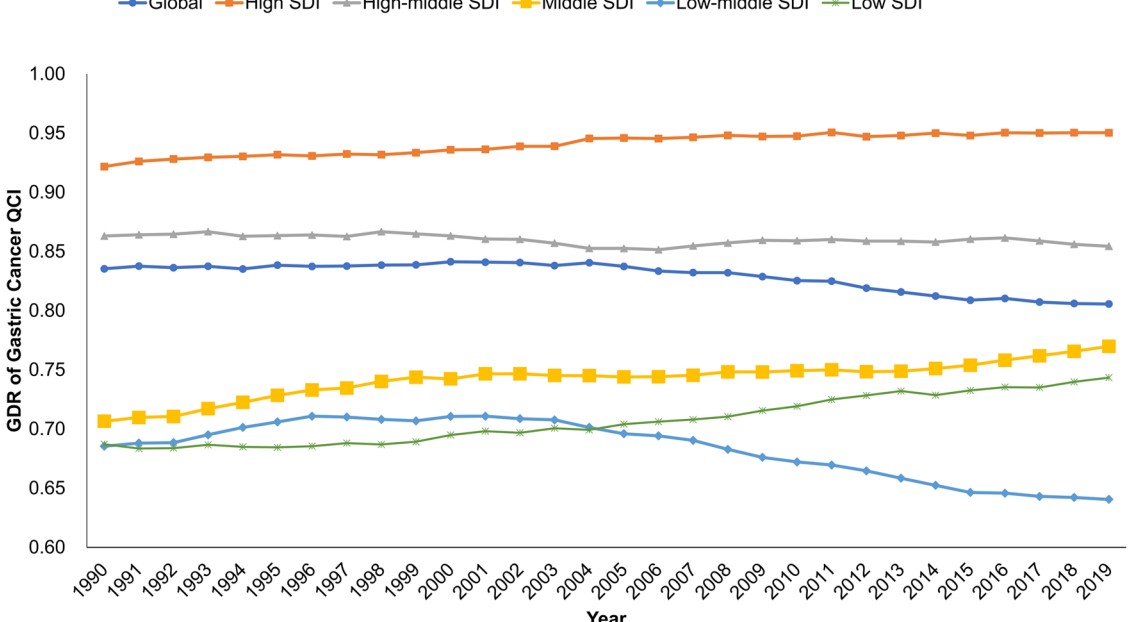

**Fig. 1 | Time trend of GDR of gastric cancer QCI, from 1990 to 2019, by SDI regions.** The GDR was defined as the ratio of the QCI score in women divided by that in men, with a GDR < 1 indicating better gastric cancer care in men compared with women. The temporal trend of GDR was displayed by country-level SDI groups, and the SDI was a comprehensive indicator based on the education level, per capita income, and total fertility rate of individuals under the age of 25 years, which measured the overall development scale of a country. GDR gender difference ratio, QCI quality of cancer care, SDI sociodemographic index.

## Results

### Gender-specific gastric cancer burden and QCI

Table 1 illustrates the gastric cancer QCI in 1990 and 2019 by gender. Globally in 2019, the gastric cancer QCI was 54.14 for females, and 67.20 for males. The estimated annual percentage change was 0.718 for females and 0.948 for males, from 1990 to 2019. Age-specific gastric cancer QCI was positively associated with social development status. Gastric cancer QCI was 86.00, 58.90, 49.94, 21.36, 13.31 for females in the high, high-middle, middle, low-middle, and low SDI regions respectively, and the corresponding QCI was 90.50, 68.93, 64.88, 33.35, 17.90 for females, respectively. From 1990 to 2019, high-middle and middle SDI regions were of the highest increment in gastric cancer QCI, with an EAPC of 3.17 and 3.88 for females and 3.21 and 3.65 for males.

### Gender disparity of gastric cancer QCI

Globally in 2019, the age-standardized death rate of gastric cancer reached 7.92 (95% uncertainty interval, UI: 7.07-8.76) per 100,000 population, and the age-standardized DALY reached 178.25 (95% UI: 160.52-196.94) per 100,000 population for females, and 16.59 (95% UI: 14.80-18.34) and 368.85 (95% UI: 328.19-410.33) for males, separately (Supplementary Table 2). The GDR of QCI of gastric cancer was 0.81 in 2019, indicating that males

received better gastric cancer care than females did. The GDR was below 1 across all SDI regions, although in regions with higher SDI, the GDR was closer to 1. This phenomenon indicated that the discrepancy between men and women in QCI of gastric cancer was the least significant in regions with high SDI. From 1990 to 2019, the GDR decreased from 0.84 to 0.81 (Fig. 1), reflecting a widening gap between genders. The GDR in the regions with low and middle SDIs improved greatly, while gender differences in the regions with a low-middle SDI showed a widening trend.

At the national level, males had better QCIs than females in most countries (Fig. 2); the five countries with the lowest GDRs were Kenya (0.30), Afghanistan (0.58), Peru (0.62), Somalia (0.63), and Bolivia (0.64). There are also countries where the QCI of women was higher than that of men, such as Lithuania (1.59), Latvia (1.55), Belarus (1.49), the Republic of Moldova (1.40), and Estonia (1.40). Age trend of GDR of gastric cancer QCI was displayed in Supplementary Fig. 1.

### Access to radiotherapy and QCI

Table 2 reported the estimated association of radiotherapy unit density, number of medical doctors with the gender-specific gastric cancer QCI. In males, univariable model suggested higher radiotherapy units density ($\beta = 0.27$, 95% CI = 0.24-0.31) was positively associated with higher gastric

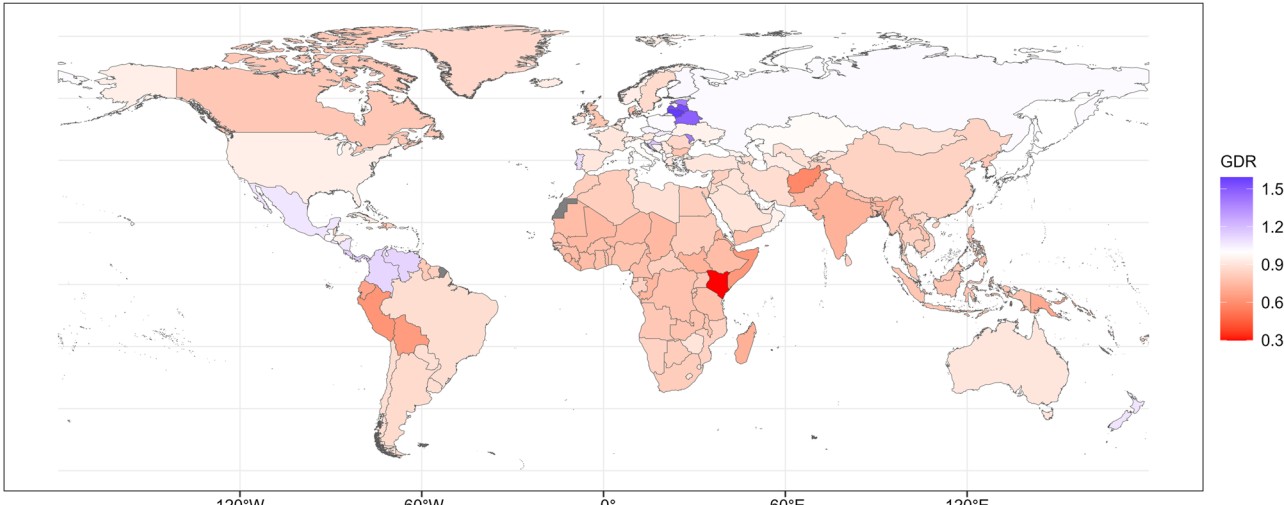

**Fig. 2 | Global map of GDR of gastric cancer QCI in 2019.** QCI index was constructed using principal component analysis and higher QCI illustrated better cancer care for gastric cancer patients. The GDR was defined as the ratio of the QCI score in women divided by that in men, with a GDR < 1 indicating better gastric cancer care in men compared with women. This figure displayed the global map of GDR of gastric cancer QCI at the country level in 2019, while the color red suggested that men had better QCI than women, and green showed a reverse association. GDR gender difference ratio, QCI quality of cancer care.

**Table 2 | Association analysis of radiotherapy unit density with gender-specific QCI and GDR**

| | QCI in males | | QCI in females | | GDR of QCI | |
|---|---|---|---|---|---|---|
| | Coefficient (95% CI) | P value | Coefficient (95% CI) | P value | Coefficient (95% CI) | P value |
| Model 1[a] | 0.27 (0.24, 0.31) | <0.001 | 0.33 (0.30, 0.37) | <0.001 | −0.04 (−0.06, −0.03) | <0.001 |
| Model 2[b] | 0.11 (0.05, 0.17) | 0.001 | 0.16 (0.09, 0.23) | <0.001 | −0.04 (−0.07, −0.01) | 0.007 |
| Model 3[c] | 0.11 (0.04, 0.17) | 0.001 | 0.15 (0.08, 0.22) | <0.001 | −0.04 (−0.07, −0.01) | 0.017 |

CI confidential interval, GDR gender difference ratio, QCI quality of cancer care.
[a]Model 1 was the univariable logistic regression model.
[b]Model 2 added the universal health coverage level and social development level defined using the sociodemographic index (SDI).as adjusting variables.
[c]Model 3 further added health workforce (number of medical doctors per 100,000 population) and infrastructure (hospital beds per 10 000 population) as covariates.

cancer QCI. The association between radiotherapy units density and gastric cancer QCI remained significant after adjusting the SDI level and UHC level for health service, that every doubling radiotherapy units per million population was associated with 0.11 (95% CI: 0.05-0.17) increasing in log-transformed value of gastric cancer QCI, equivalent to 1.12 increasing in QCI value. Estimation in model 3 reported a significant association (0.11, 95% CI: 0.04-0.17) after further controlling the health workforce and infrastructure. Similar and stronger association was observed for females that the estimated coefficient reached 0.33 (95% CI: 0.30-0.37) in univariable model and 0.16 (95% CI: 0.09-0.23) and 0.15 (95% CI: 0.08-0.22) in multivariable model for radiotherapy unit density.

When using GDR of gastric cancer QCI as the outcome variable, results from the regression model suggested that radiotherapy unit was reversely associated with the GDR, with the estimated coefficient reaching −0.04 (95% CI: −0.06, −0.03) in univariable model, and -0.04 (95% CI: −0.07, −0.01) in multivariable models after adjusting SDI level and UHC coverage, and −0.04 (95% CI: −0.07, −0.01) after further adjusting health workforce and hospital facilities.

## Discussion
Gastric cancer is one of the leading causes of death among all types of cancer. It is estimated that there will be 1,738,190 new cases and 1,303,138 deaths globally in 2040, resulting in health and economic burdens[1]. Providing sustainable high-quality cancer care is a key strategy to improve quality of life for cancer patients. In this study, we measured the quality of care for gastric cancer and assessed the gender disparity. The role of access to radiotherapy in promoting gastric cancer care quality and equality was further explored. Males tended to receive better quality of gastric cancer care than females, and this gender disparity was seen to widen in middle-low SDI regions. Remarkably, a positive correlation emerged between the density of radiotherapy facilities and an elevated QCI for both genders, and culminating in an ameliorated gender imbalance.

The prevention and control strategies for gastric cancer have been established and supported with epidemiological evidence[20], where providing qualified cancer care was one of the key pillars in comprehensive approaches. Risk factors such as Helicobacter pylori infection have been shown to significantly increase the likelihood of developing stomach cancer[21,22]. The International Agency for Research on Cancer (IARC) has recognized the eradication of H. pylori as one of the global strategies to prevent gastric cancer[23]. Endoscopy has proven to be cost-effective in reducing gastric cancer mortality in high-incidence areas like East Asia[24]. Providing high-quality treatment and care for diagnosed patients, including the establishment of multidisciplinary teams, can also improve patient outcomes and quality of life[25].

This study found that men receive higher quality gastric cancer care than women, and the gender gap is widening in regions with a low-middle development level. This difference may be due to biological or sociological factors. Females are more often diagnosed with a diffuse-type gastric adenocarcinoma (GAC) and signet cell ring GAC[26,27], and are more likely to have microsatellite instability (MSI); such biological differences may lead to poorer survival and affect the QCI levels measured based on health outcomes in this study. From a sociological perspective, studies have shown that women are more likely to be undertreated for cancer, and the proportion of female stomach cancer patients receiving systemic treatment is lower than that of men[28], which also reduces the quality of care for women.

Genomic characterization has enhanced the classification and prognosis of gastric cancer. For instance, MSI stands out among the genomic markers, characterized by short, repetitive DNA sequences randomly dispersed throughout the genome. The prevalence of MSI is notably lower among Asians (<10%)[29] compared to Western populations (22%)[30]. Gastric cancers with MSI typically associated with older age, female gender, distal stomach location, and a reduced number of lymph-node metastases[31–33]. Recent evidence suggests a favorable prognostic significance associated with microsatellite instability status[29,34]. The heightened immunogenicity and widespread expression of immune-checkpoint ligands render the microsatellite instability subtype particularly susceptible to immunotherapeutic interventions, such as treatment with anti-PD-L1 and anti-CTLA4 antibodies[32,35,36]. Tailored immunotherapeutic trials hold promise in facilitating the identification of precision treatment strategies[37].

The findings of the analysis revealed a discernible correlation between access to radiotherapy units and an elevated standard of gastric cancer care in both genders, accompanied by a reduction in gender-based disparities. Notably, this relationship retained its significance even following adjustments for confounding variables including social development levels, characteristics of the health system, the health workforce, and health facilities. Radiotherapy is among the most cost-effective, efficient and widely-used cancer treatments, and may be considered as a treatment option for an estimated half of cancer patients. In the context of gastric cancer, gastrectomy assumed the pivotal role of a curative cornerstone for localized cases, as underscored by its status as the mainstay of treatment. Moreover, a systematic review yielded compelling evidence of a statistically significant survival advantage over a 5-year period following the inclusion of radiotherapy in the therapeutic regimen of patients diagnosed with resectable gastric cancer[38]. This observation reinforces the enduring pertinence of radiotherapy as a canonical element in the management of resectable gastric cancer[39]. Despite being a critical component of cancer care, however, worldwide access to radiotherapy is still inadequate, particularly in lower-income countries. To reinvigorate efforts to address this problem, WHO joined forces with the International Atomic Energy Agency and launched the Rays of Hope Initiative in 2022, to support Member States in providing their people access to diagnosis and treatment of cancer using radiation medicine[40]. Findings from this study confirmed the association between access to radiotherapy units and improved gastric cancer QCI, as well as narrowed gender disparity. Notably, these insights supported the action of supplying essential radiation medicine to populations and countries vulnerable to these afflictions.

This study has several strengths. We quantify the gender disparity in accessing gastric cancer care, and exploring health system factors to inform further actions. This study also has limitations. The index of quality of care was constructed from the disease burden, the constructed QCI index may be biased due to underdiagnosis in areas with difficult access to health services, and the estimated EAPC indicator may be biased with a nonlinear temporal trend, then the trend and disparity of gastric cancer care should be with caution in interpretation. This article only identified the association between access to radiotherapy units and quality of gastric cancer care, further studies could expand the cancer types especially cancers more sensitive to radiotherapy to reveal the imperative of fortifying radiotherapy infrastructure within areas and populations in greatest need.

## Conclusion

In summary, females were vulnerable and hence should be given more attention to gastric cancer management and care. The association between robust radiotherapy access, improved gastric cancer QCI, and reduced gender-based disparities spotlighted the imperative of fortifying radiotherapy infrastructure within areas and populations in greatest need.

## Data availability

The source data is located in Supplementary Data 1.

## Code availability

The codes used for our analyses will be available upon reasonable request from the corresponding author Y.J. Requests should be directed to Dr. Yinzi Jin at yzjin@bjmu.edu.cn.

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

## Acknowledgements

This study was funded by the Bill and Melinda Gates Foundation (No. INV-0045085). The study sponsor had no role in the study design, data analysis and interpretation of data, the writing of the manuscript, or the decision to submit the paper for publication.

## Author contributions

All authors helped develop the study concept and design. M.W. and K.H. contributed to the data collection, and statistical analysis. M.W. drafted the manuscript. X.F. and J.W. provided technical support for the data interpretation. Y.J. and Z.Z. provided overall guidance and critical revision. All authors revised the manuscript and approved the final version.

## Competing interests

The authors declare no competing interests.
