## [Peer Review File · Communications Medicine]

Reviewers' comments:

Reviewer #1 (Remarks to the Author):

Comments to the author

In this paper, Minmin Wang and colleagues assessed the disparity of gastric cancer care across genders and geography along with the association with the radiotherapy unit density, which could fortify radiotherapy infrastructure in greatest need and promote gender equality. However, I have some suggestions to help improve the manuscript.

1. The following paper has summarized the Quality of Care Index (QCI) of Gastric Cancer in the world. The authors need to summarize the latest research progress on this topic, and clearly extract the novelty of this study.

Fattahi N, Ghanbari A, Djalalinia S, Rezaei N, Mohammadi E, Azadnajafabad S, Abbasi-Kangevari M, Aryannejad A, Aminorroaya A, Rezaei N, Azmin M, Ramezani R, Jafari F, Aghili M, Farzadfar F. Global, Regional, and National Quality of Care Index (QCI) of Gastric Cancer: A Systematic Analysis for the Global Burden of Disease Study 1990-2017. *J Gastrointest Cancer*. 2023 Jun 26. doi: 10.1007/s12029-023-00950-3. Epub ahead of print. PMID: 37365424.

2. The role and importance of radiotherapy in the treatment of gastric cancer should be explained in detail in the Introduction.

3. Moreover, for the imperative of fortifying radiotherapy infrastructure within areas and populations in greatest need, and promote gender equality policy, the manuscript should include all cancer types or tumors that are more sensitive to radiation therapy.

4. The manuscript included the global health estimates analysis; therefore, the writing should comply with the GATHER checklist. The checklist need to be supplied.

5. The methods section is not written professionally. Although this study is a secondary analysis based on the GBD study and GHO datasets, some essential information, such as number and distribution of original sources, and data integration processes, need to be delineated to get an overall assessment of original evidence. Moreover, the related references on statistical methods have been cited in the methods section, such as EAPC and GDR calculation.

6. When exploring the association of radiotherapy unit density, number of medical doctors with the gender-specific gastric cancer QCI, the authors should represent the distribution information for these variables. I am not sure if the specific data used is at the national or regional level.

7. The EAPC in QCI between 1990 and 2019 cannot be landscape characteristics, especially since the trends may be not linear over time. The authors should report the shortage of the average indicators of long-term trends over the past few decades.

8. Discussion: being this is an ecologic study, the overall quality of the results depends on the overall quality of the data; such as the disease diagnosis and measures of radiotherapy infrastructure over time, the disease diagnosis may be underdiagnosed in locations characterized by difficult access to health services, particularly in deprived economic groups of the general population, you should discuss such a topic.

9. The author should provide a detailed description of the shortcomings of the manuscript.

Reviewer #2 (Remarks to the Author):

In this article the authors retrieved the gastric cancer disease burden data from Global Burden of Disease year 2019 and used PCA to construct a quality of care index (QCI). The focus was to look at the disparity of gastric cancer care across genders and to obtain the association of access to radiotherapy with gastric cancer care as well as gender disparity. The results indicate that in the study period, males tended to receive better quality of gastric cancer care than females. This gender disparity was more prominent in middle-low socio-development regions. The data also shows a positive correlation between the density of radiotherapy facilities and an elevated QCI for both genders. The use of The Global Burden of Disease (GBD) database covering 204 countries and 21 regions is a strength. Overall, the results are certainly interesting, the introduction and discussion covers the topic in great detail and the data is clearly presented.

Specific comments:

The limitation of the study is too brief. The authors may well enhance the discussion related to limitations related to this type of study.

The MSI high disease rates in women could be further discussed and the authors may also include a paragraph on unique targeted treatments for such patient population to enhance the discussion session

Legends to figures should be enhanced

The article should be carefully checked for grammar and typographical errors.

Reviewer #1 (Remarks to the Author):

In this paper, Minmin Wang and colleagues assessed the disparity of gastric cancer care across genders and geography along with the association with the radiotherapy unit density, which could fortify radiotherapy infrastructure in greatest need and promote gender equality. However, I have some suggestions to help improve the manuscript.

1. The following paper has summarized the Quality of Care Index (QCI) of Gastric Cancer in the world. The authors need to summarize the latest research progress on this topic, and clearly extract the novelty of this study.

Fattahi N, Ghanbari A, Djalalinia S, Rezaei N, Mohammadi E, Azadnajafabad S, Abbasi-Kangevari M, Aryannejad A, Aminorroaya A, Rezaei N, Azmin M, Ramezani R, Jafari F, Aghili M, Farzadfar F. Global, Regional, and National Quality of Care Index (QCI) of Gastric Cancer: A Systematic Analysis for the Global Burden of Disease Study 1990-2017. *J Gastrointest Cancer*. 2023 Jun 26. doi: 10.1007/s12029-023-00950-3. Epub ahead of print. PMID: 37365424.

Authors' Response: We thank the comments of the reviewer. We have noticed this article describing the global quality of gastric cancer care by using the global burden of disease 2017 and we have cited this article in Introduction Section to illustrate the latest research progress on this topic. As we highlighted in the Introduction, this article provided evidence on the great disparity of gastric cancer care across regions and countries, while it did not deeply explore the potential reasons behind the disparity as well as the improvement strategies. In this study, we further identify the association between access to radiotherapy and quality of gastric cancer care and explain the mechanism behind the disparity of gastric cancer care

across SDI levels and genders. To better explain the novelty of this study, we have added the related sentence in the Introduction Section as “Previous study has estimated the global quality of gastric cancer care, and reported great disparity across regions, countries and genders, which limited the progress to improve the quality of life in gastric cancer patients as well as to improve the global health equity. However, neither the mechanism behind the disparity of quality of gastric cancer care across regions, countries and genders was fully understood, nor the improving strategies were informed. Access to essential treatment infrastructure, especially the radiotherapy units, could impact the quality of gastric cancer care and reduce the gender-based disparity...This study aimed to explore the mechanism of the disparity of gastric cancer care across regions, countries, and genders, by conducting an association analysis to identify the role of health technologies and health workforce in promoting gastric cancer quality and equality.” (Page 4, Line 12- Page 5, Line 14, in clean copy)

2. The role and importance of radiotherapy in the treatment of gastric cancer should be explained in detail in the Introduction.

Authors’ Response: We thank the comments of the reviewer. The essential role of radiotherapy in treatment of gastric cancer was the fundamental of the association analysis. We have added the related sentence regarding the role and importance of radiotherapy in treatment of gastric cancer into the Introduction Section as “Most patients with gastric cancer are locally advanced and require multimodal treatment, in which radiotherapy is emphasized. It is generally accepted that adjuvant and neoadjuvant therapies, such as adjuvant chemoradiotherapy and preoperative chemoradiotherapy, can improve the disease-free and overall survival of patients with locally advanced gastric cancer who have undergone

adequate and complete surgical resection. With these therapies, the 5-year overall survival rate can be increased by 10-15%” (Page 5, Lines 3-9, in clean copy).

3. Moreover, for the imperative of fortifying radiotherapy infrastructure within areas and populations in greatest need, and promote gender equality policy, the manuscript should include all cancer types or tumors that are more sensitive to radiation therapy.

Authors’ Response: We thank the comments of the reviewer. In this article, we are aimed to explore the association between radiotherapy units and the gastric cancer care. Inspired by the results identified in the article, as well as the suggestion raised by the reviewer, we are planning to conduct a further study to explore the impact of radiotherapy access with all cancer care and by cancer types. We also added the related discussion in the limitation section as “This article only identified the association between access to radiotherapy units and quality of gastric cancer care, further studies could expand the cancer types especially cancers more sensitive to radiotherapy to reveal the imperative of fortifying radiotherapy infrastructure within areas and populations in greatest need.” (Page 16 Lines 11-15, in clean copy).

4. The manuscript included the global health estimates analysis; therefore, the writing should comply with the GATHER checklist. The checklist need to be supplied.

Authors’ Response: We thank the comments of the reviewer. The GATHER checklist has been applied and added in the supplemental material.

5. The methods section is not written professionally. Although this study is a secondary analysis based on the GBD study and GHO datasets, some essential information, such as number and distribution of original sources, and data integration processes, need to be

delineated to get an overall assessment of original evidence. Moreover, the related references on statistical methods have been cited in the methods section, such as EAPC and GDR calculation.

Authors' Response: We thank the comments of the reviewer. In methods section, we further added the methodology used in GBD study and GHO datasets in the revised manuscript as “Briefly, the GBD estimation process is based on identifying multiple relevant data sources for each disease including censuses, household surveys, civil registration and vital statistics, disease registries, health service use, air pollution monitors, satellite imaging, disease notifications, and other sources. Cause-specific death rates and cause fractions were calculated using the Cause of Death Ensemble model and spatiotemporal Gaussian process regression. Cause-specific deaths were adjusted to match the total all-cause deaths calculated as part of the GBD population, fertility, and mortality estimates. Deaths were multiplied by standard life expectancy at each age to calculate years of life lost (YLLs). A Bayesian meta-regression modelling tool, DisMod-MR 2.1, was used to ensure consistency between incidence, prevalence, remission, excess mortality, and cause-specific mortality for most causes. Prevalence estimates were multiplied by disability weights for mutually exclusive sequelae of diseases and injuries to calculate years lived with disability (YLDs).” (Page 6, Lines 6-19, in clean copy) and “In 2010, WHO launched a country survey on medical devices that allowed to identify the status of high cost medical devices in the Member States, including radiotherapy equipment, both linear accelerators and Cobalt-60. Similar survey was conducted in 2020-2021 update by collecting information directly from country focal points from ministries of health” (Page 7, Lines 10-14, in clean copy). And the reference for EPAC and GDR calculation was also added into revised manuscript.

6. When exploring the association of radiotherapy unit density, number of medical doctors

with the gender-specific gastric cancer QCI, the authors should represent the distribution information for these variables. I am not sure if the specific data used is at the national or regional level.

Authors' Response: We thank the comments of the reviewer. The specific data used in the association analysis, such as the radiotherapy unit density, number of medical doctors with the gender-specific gastric cancer QCI, were all estimated at country level since they were collected from country focal points from ministries of health through WHO surveys. We further detailed illustrated the data source of the GBD study as well as the methodology of GHO database to avoid misunderstanding, as “In 2010, WHO launched a country survey on medical devices that allowed to identify the status of high cost medical devices in the Member States, including radiotherapy equipment, both linear accelerators and Cobalt-60. Similar survey was conducted in 2020-2021 update by collecting information directly from country focal points from ministries of health.” (Page 7, Lines 10-14, in clean copy) in the revised manuscript.

7. The EAPC in QCI between 1990 and 2019 cannot be landscape characteristics, especially since the trends may be not linear over time. The authors should report the shortage of the average indicators of long-term trends over the past few decades.

Authors' Response: We thank the comments of the reviewer. We admitted that EAPC would be biased if the temporal trend may be not linear over time. We have added the related shortage of the average indicators in the limitation section as “The index of quality of care was constructed from the disease burden, the constructed QCI index may be biased due to underdiagnosis in areas with difficult access to health services, and the estimated EAPC indicator may be biased with a nonlinear temporal trend, then the trend and disparity of

gastric cancer care should be with caution in interpretation” (Page 16, Line 7-11, in clean copy).

8. Discussion: being this is an ecologic study, the overall quality of the results depends on the overall quality of the data; such as the disease diagnosis and measures of radiotherapy infrastructure over time, the disease diagnosis may be underdiagnosed in locations characterized by difficult access to health services, particularly in deprived economic groups of the general population, you should discuss such a topic.

Authors’ Response: We thank the comments of the reviewer. We agreed with the reviewer that disease diagnosis and measures of radiotherapy infrastructure was significant in ensuring the quality of the research. For measures of radiotherapy infrastructure, we have detailed explained the data source of this variable, which was collected from country focal points from ministries of health, through a WHO-led country survey. Among 172 countries included into current study, the information of radiotherapy infrastructure in 140 countries were updated during 2018-2020, indicating the latest availability of radiotherapy infrastructure at country level. We admitted that underdiagnosis of gastric cancer in areas with difficult access to health services may bias the estimation of gastric cancer care and we have discussed the potential impact in the limitation section as “The index of quality of care were constructed from the disease burden, the constructed QCI index may be biased due to underdiagnosis in areas with difficult access to health services, and the estimated EAPC indicator may be biased with nonlinear temporal trend, then the trend and disparity of gastric cancer care should be with caution in interpretation” (Page 16, Line 7-11, in clean copy).

9. The author should provide a detailed description of the shortcomings of the manuscript.

Authors' Response: We thank the comments of the reviewer. Based on the reviewers' comments and suggestions, the limitations of the article have been broadly discussed and evaluated in the revised manuscript as "This study also has limitations. The index of quality of care were constructed from the disease burden, the constructed QCI index may be biased due to underdiagnosis in areas with difficult access to health services, and the estimated EAPC indicator may be biased with nonlinear temporal trend, then the trend and disparity of gastric cancer care should be with caution in interpretation. This article only identified the association between access to radiotherapy units and quality of gastric cancer care, further studies could expand the cancer types especially cancers more sensitive to radiotherapy to reveal the imperative of fortifying radiotherapy infrastructure within areas and populations in greatest need." (Page 16, Lines 7-15, in clean copy).

Reviewer #2 (Remarks to the Author):

In this article the authors retrieved the gastric cancer disease burden data from Global Burden of Disease year 2019 and used PCA to construct a quality of care index (QCI). The focus was to look at the disparity of gastric cancer care across genders and to obtain the association of access to radiotherapy with gastric cancer care as well as gender disparity. The results indicate that in the study period, males tended to receive better quality of gastric cancer care than females. This gender disparity was more prominent in middle-low socio-development regions. The data also shows a positive correlation between the density of radiotherapy facilities and an elevated QCI for both genders. The use of The Global Burden of Disease (GBD) database covering 204 countries and 21 regions is a strength. Overall, the results are certainly interesting, the introduction and discussion covers the topic in great detail and the data is clearly presented.

Specific comments:

The limitation of the study is too brief. The authors may well enhance the discussion related to limitations related to this type of study.

Authors' Response: We thank the comments of the reviewer. Based on the reviewers' comments and suggestions, the limitations of the article have been broadly discussed and evaluated in the revised manuscript as "This study also has limitations. The index of quality of care were constructed from the disease burden, the constructed QCI index may be biased due to underdiagnosis in areas with difficult access to health services, and the estimated EAPC indicator may be biased with nonlinear temporal trend, then the trend and disparity of gastric cancer care should be with caution in interpretation. This article only identified the association between access to radiotherapy units and quality of gastric cancer care, further studies could expand the cancer types especially cancers more sensitive to radiotherapy to reveal the imperative of fortifying radiotherapy infrastructure within areas and populations in greatest need." (Page 16, Lines 7-15, in clean copy).

The MSI high disease rates in women could be further discussed and the authors may also include a paragraph on unique targeted treatments for such patient population to enhance the discussion session

Authors' Response: We thank the comments of the reviewer. MSI was one of the genomic characterizations of gastric cancer with clinical perspectives and treatment approaches. As suggested by the reviewer, we further added one paragraph in Discussion section focused on the unique targeted treatments for such patient population as "Genomic characterization has significantly enhanced the classification and prognosis of gastric cancer. For instance, MSI stands out among the genomic markers, characterized by short, repetitive DNA sequences

randomly dispersed throughout the genome. The prevalence of MSI is notably lower among Asians (<10%) compared to Western populations (22%). Gastric cancers with MSI typically associated with older age, female gender, distal stomach location, and a reduced number of lymph-node metastases. Recent evidence suggests a favourable prognostic significance associated with microsatellite instability status. The heightened immunogenicity and widespread expression of immune-checkpoint ligands render the microsatellite instability subtype particularly susceptible to immunotherapeutic interventions, such as treatment with anti-PD-L1 and anti-CTLA4 antibodies. Tailored immunotherapeutic trials hold promise in facilitating the identification of precision treatment strategies.” (Page 14, Line 12- Page 15, Line 2, in clean copy).

Legends to figures should be enhanced

Authors’ Response: We thank the comments of the reviewer. The figure legends have been revised as “Figure 1. Time trend of GDR of gastric cancer QCI, from 1990 to 2019, by SDI regions. The GDR was defined as the ratio of the QCI score in women divided by that in men, with a GDR <1 indicating better gastric cancer care in men compared with women. The temporal trend of GDR was displayed by country-level SDI groups, and the SDI was a comprehensive indicator based on the education level, per capita income, and total fertility rate of individuals under the age of 25 years, which measured the overall development scale of a country.” and “QCI index was constructed using principal component analysis and higher QCI illustrated better cancer care for gastric cancer patients. The GDR was defined as the ratio of the QCI score in women divided by that in men, with a GDR <1 indicating better gastric cancer care in men compared with women. This figure displayed the global map of GDR of gastric cancer QCI at the country level in 2019, while the color red suggested that men had better QCI than women, and green showed a reverse association.”.

The article should be carefully checked for grammar and typographical errors.

Authors' Response: We thank the comments of the reviewer. Related grammar and typographical errors have been carefully checked and revised throughout the manuscript.

REVIEWERS' COMMENTS:

Reviewer #1 (Remarks to the Author):

The authors have revised the manuscript focused on all queries.

Reviewer #2 (Remarks to the Author):

Comments have been addressed.